# Investigating the Risk Factors Associated with Injury Severity in Pedestrian Crashes in Santiago, Chile

**DOI:** 10.3390/ijerph191711126

**Published:** 2022-09-05

**Authors:** Angelo Rampinelli, Juan Felipe Calderón, Carola A. Blazquez, Karen Sauer-Brand, Nicolás Hamann, José Ignacio Nazif-Munoz

**Affiliations:** 1Faculty of Engineering, Universidad Andres Bello, Antonio Varas 880, Santiago 7500971, Chile; 2Unidad de Innovación Docente y Académica, Universidad Andres Bello, Quillota 980, Viña del Mar 2531015, Chile; 3Department of Engineering Sciences, Universidad Andres Bello, Quillota 980, Viña del Mar 2531015, Chile; 4Faculty of Economics and Business, Universidad Andres Bello, Fernández Concha 700, Santiago 7591538, Chile; 5Faculty of Engineering, Universidad Andres Bello, Quillota 980, Viña del Mar 2531015, Chile; 6Faculté de Médecine et des Sciences de la Santé, Université de Sherbrooke, 150, Place Charles-Le Moyne, Longueuil, QC J4K 0A8, Canada

**Keywords:** pedestrian safety, spatial analysis, traffic injury, kernel density estimation, partial proportional odds

## Abstract

Pedestrians are vulnerable road users that are directly exposed to road traffic crashes with high odds of resulting in serious injuries and fatalities. Therefore, there is a critical need to identify the risk factors associated with injury severity in pedestrian crashes to promote safe and friendly walking environments for pedestrians. This study investigates the risk factors related to pedestrian, crash, and built environment characteristics that contribute to different injury severity levels in pedestrian crashes in Santiago, Chile from a spatial and statistical perspective. First, a GIS kernel density technique was used to identify spatial clusters with high concentrations of pedestrian crash fatalities and severe injuries. Subsequently, partial proportional odds models were developed using the crash dataset for the whole city and the identified spatial clusters to examine and compare the risk factors that significantly affect pedestrian crash injury severity. The model results reveal higher increases in the fatality probability within the spatial clusters for statistically significant contributing factors related to drunk driving, traffic signage disobedience, and imprudence of the pedestrian. The findings may be utilized in the development and implementation of effective public policies and preventive measures to help improve pedestrian safety in Santiago.

## 1. Introduction

Walking is a common human activity that provides health and economic benefits, and helps reduce traffic congestion, air pollution, carbon emissions, and energy consumption [1]. Thus, walking contributes to a better quality of life and sustainability. However, pedestrians are vulnerable road users with a high likelihood of being injured or killed when involved in traffic crashes. Pedestrians are directly exposed to the impact of these crashes, and thus, the fatality risk of pedestrians is higher than vehicle occupants [2,3].

According to the World Health Organization (WHO), over 300,000 pedestrian fatalities were reported worldwide in 2018, accounting for 26% of all traffic deaths, while millions of victims are injured in pedestrian crashes every year [4]. In recent years, pedestrian safety has been a relevant research area as increased efforts have focused on promoting active travel such as walking in the population [5]. Chile has the highest walking rate in Latin America with an average of 5204 steps per day, which has helped improve public health campaigns to fight obesity [6]. However, Chile has the highest fatality rate of pedestrians in traffic crashes among other Organization for Economic Co-operation and Development (OECD) country members with 3.6 road deaths per 10,000 motorized vehicles. Pedestrians represent the largest share of casualties with 34% among all road user groups, and approximately 20% of all severely injured victims due to road traffic crashes nationwide [7].

Santiago is the capital and largest city of Chile with over 700 km^2^ of extension and a population of 7.1 million inhabitants, representing approximately 40% of the total population of Chile [8]. Walking is the major mode of transportation in Santiago [9,10]. Over six million walking trips are generated every day in the city [11]. People walk on average 0.67 km every day to commute to work, and 19% of these inhabitants walk more than 1 km to reach a specific destination [12]. Although sidewalks are abundant, the majority of major road infrastructure favors automobile users, making pedestrian infrastructure in some places inhospitable and disrupted [13]. Thus, pedestrian safety is a critical concern. Nearly 30% of all pedestrian crashes reported nationwide occur in Santiago, yielding more than 40% of all crash fatalities and 27% of all crash severely injured victims that arise in Chile [14]. Therefore, there is an imperative need to investigate the risk factors that contribute to pedestrian crash severity in Santiago for implementing preventive measures that will have the most significant impact on pedestrian safety.

## 2. Literature Review

Different risk factors (such as demographics, crash, and built environment characteristics) that affect pedestrian crash injury severity and casualty have been identified by several studies [15,16,17,18,19,20,21,22,23,24,25,26,27,28]. From the demographic perspective, males and the elderly are found to be more prone to incur serious injuries and fatalities than other age groups when involved in pedestrian crashes [16,17,29,30]. Other studies have concluded that more severe pedestrian injuries and fatalities are associated with temporal factors, such as time of day, day of the week, and season [15,19,31,32]. For instance, mechanisms that explain increases in crash severity may be the lack of visibility at night-time. Among different contributing factors of pedestrian crashes, the influence of alcohol consumption has shown to increase injury severity in pedestrian crashes [26,33,34]; at-fault drivers cause fatalities and serious injuries when not yielding right-of-way to pedestrians at intersections and crosswalks [35,36]; and the irresponsible or careless behavior of pedestrians have proven to be associated with more serious crash injuries [37,38]. In terms of built environment factors, high rates of pedestrian crash casualties tend to occur in neighborhoods of lower socioeconomic status (SES), meaning that pedestrians that walk in deprived neighborhoods have higher probability of being killed or seriously injured in a crash than in affluent neighborhoods [5,39,40,41,42]. Additionally, the odds of crashes that result in injured or killed pedestrians increase in commercial and residential areas particularly due to large flow of pedestrians and vehicles, and high intersection density [15,43]. In addition, studies suggest that there is a higher risk of crashes with fatal and severe injury outcomes in areas with higher population density, in which pedestrians are more exposed to crashes [5,16]. Public transportation accessibility has been associated with pedestrian crash injury severity, suggesting the need of safety zones near bus stops and subway stations to protect pedestrians [15,23,44,45]. Finally, studies have concluded that pedestrians suffer more severe injuries at road intersections due complex traffic operations inherent to such intersections [3,21,36,46,47].

Econometric methods have been widely employed to analyze the effect of risk factors on injury severity and fatalities in pedestrian crashes, such as multinomial logit models [26,48,49], mixed logit models [19,44,50], and ordered logit or probit models [16,47,51]. However, multinomial logit models and mixed logit models assume that all injury severity levels are non-ordered without considering the inherent hierarchical nature of crash injury severities [19], whereas the ordered logit or probit models are ordered-response models that account for the inherent ordered nature of the dependent variable (i.e., levels of injury severity). In addition, multinomial logit models are less parsimonious and more difficult to interpret than the ordered response models [52]. Ordered logit or probit models follow the parallel-lines assumption, in which parameter estimates are equal and constant across different levels of the dependent variable. This is usually not the case with crash injury severity outcomes since explanatory variables may have the same or dissimilar effects on different injury severity levels [30]. The partial proportional odds (PPO) model relaxes the parallel-lines assumption, allowing certain variables to violate the parallel-lines assumption while other variables are constrained to this assumption [52], and thus, the model results are more correct and complete than ordered logit or probit models.

Studies have developed PPO models to study risk factors affecting crash injury severity concerning at-fault drivers and not at-fault drivers [53], heavy trucks [27,54,55], bicycles [56], work zones [57], rural highways [58], and wrong-way driving [59], among others. PPO models have also proven to outperform other statistical models when examining the effect of risk factors of pedestrian injuries and fatalities in traffic crashes [18,24,25,60]. Thus, the PPO model is used in this study to explore the main risk factors that significantly affect severity injury of pedestrian crashes, as in studies by Lin and Fan [17], Pour et al. [32], and Lu et al. [46].

In traffic crash analysis, kernel density estimation (KDE) is a commonly used spatial analytical tool to identify hazardous road locations. KDE adds a significant value to the statistical methods by revealing high-risk areas for pedestrians, in which there is an increased likelihood of crashes based on spatial dependency [31,61,62], and where authorities need to focus on specific countermeasures [43,63]. Thus, studies have combined KDE and statistical methods for analyzing injury severity in clusters with high density of pedestrian crashes, which are summarized in Table 1 [15,22,43,62,63,64,65].

To the best of our knowledge, this is the first study that contributes to the literature by investigating the risk factors associated with pedestrian crash injury severity in Chile from a spatial and statistical perspective. First, a KDE method is employed in a GIS environment to identify spatial clusters with high likelihood of pedestrian crashes that resulted in serious injuries and fatalities. Subsequently, the PPO models are developed to understand different risk factors related to the pedestrian, crash, and built environment characteristics that significantly affect injury severity levels of pedestrian crashes within the identified spatial clusters, which are then compared with the PPO model results for the whole pedestrian crash dataset in Santiago. Therefore, this study contributes to better inform current debates regarding appropriate countermeasures to enhance pedestrian safety.

## 3. Data

During the 2012–2016 period, the Chilean police reported 4874 pedestrian crashes with different injury severity levels that occurred in Santiago, of which 4216 (86.5%) were successfully geocoded [14], as shown in Figure 1. The remaining crash locations were not geocoded because the addresses or intersections were incomplete, incorrect, or non-existent. The pedestrian crash database that includes demographics and crash characteristics was requested from the Chilean National Road Safety Commission (CONASET) through the Transparency Law [66]. The built environment characteristics, such as socioeconomic status, land use, and access to public transportation, are other risk factors considered in the injury severity analysis that were provided by private and public entities (e.g., National Statistics Institute, Spatial Data Infrastructure-City Observatory, and Center for Sustainable Urban Development). Among the built environment characteristics considered in this study, a risk factor related to the amount of exposed population to pedestrian crashes was computed using the number of residents that reside in the vicinity of pedestrian crashes using the 2017 Census [8], which is a surrogate factor to the pedestrian volume counts near these crashes that actually account for the quantity of walking people. Based on the provided crash data, a three-point ordinal scale from lowest to highest was used to define injury severity level (1 = less serious injury, 2 = serious injury, 3 = fatal injury, denoting B, A, K, respectively, in the KABCO scale established by the United States Federal Highway Administration). Crashes with at least one injured pedestrian or one pedestrian that died within 24 h of the crash were considered in the analyses.

Table 2 shows the descriptive statistics with the frequency and percentages of pedestrian crashes for each analyzed risk factor and the percentages for the three injury severity levels examined in this study. This table indicates that pedestrian crash occurrence yielded less seriously injured pedestrians (22.2%), seriously injured pedestrians (64.8%), and fatal severities (13.0%). Approximately 50% of all persons involved in pedestrian crashes are adults (25–65 years old) and the proportion of males involved in pedestrian crashes is higher than females (57% males vs. 43% females). For the relative location factor, pedestrian crashes that occurred along straight road sections represent the highest share with 42.1%, followed by intersections with functioning traffic lights with 33.7%.

The most critical time is during night-time due to the higher number of pedestrian crashes and higher proportion of fatal crashes compared to other times of the day. Higher proportions of less seriously and seriously injured victims are perceived during holidays, while a higher share of fatal pedestrian crashes occur during weekends. A larger number of pedestrian crashes arose in the winter season with a higher share of less seriously injured victims. The imprudence of the pedestrian (e.g., pedestrian crosses road surprisingly or carelessly, jaywalking) and the imprudence of the driver (e.g., not yielding right of way to pedestrians, improper turns) account for 30.4% and 34.1%, respectively, of all contributing factors, while disobeying traffic signals has the highest share of fatal injury with 26.9%, followed by driving and pedestrians being under the influence of alcohol with 24.3% and 23.2%, respectively.

Regarding built environment factors, approximately 64.7% of the pedestrian crashes occur in medium and medium-low SES zones of the city, but a higher proportion of fatal crashes is observed in low SES neighborhoods (20.0%). In addition, pedestrian crashes have a tendency of occurring in commercial (33.9%) and residential areas (32.8%), resulting in similar distributions of injury severity across different land uses. Although most pedestrian crashes occur where the exposed population is low (<500 inhabitants), similar proportions are observed in the pedestrian crash injuries among locations with different levels of population exposure. Approximately 83% and 51.2% of the pedestrian crashes occur in the vicinity of bus stops and intersections, respectively, while more than half of the crashes occurred away from subway stations and traffic lights.

## 4. Methods

As aforementioned, in this study, the spatial phenomenon is incorporated in the statistical analysis through the use of KDE to investigate the influence of risk factors associated with injury severity in pedestrian crashes in critical areas (dangerous locations) of Santiago. Subsequently, these results are compared with significant factors identified for pedestrian crashes that occurred in the whole city.

### 4.1. Kernel Density Estimation (KDE)

KDE is used in this study to identify risk areas with high density of pedestrian crashes in Santiago. KDE is a non-parametric method that first yields a symmetrical surface over each point feature, secondly assesses the distance from the point to a reference location based on a mathematical function, and finally adds the value for all the surfaces for that reference location [67]. In this study, Equation (1) is used to compute the estimated density value of pedestrian crashes at location (*x*,*y*) by providing a magnitude per unit of area (e.g., the number of pedestrian crashes per km^2^).
(1)$$f\left(x,y\right)=\left(\frac{1}{nh^{2}}\right)\overset{n}{\underset{i=1}{\sum }}K\left(\begin{array}{c} d_{i}\\ h \end{array}\right)$$
where *n* is the total number of locations, *h* is the bandwidth or smoothing parameter, *K* is the kernel function, and *d_i_* is the distance between location (*x*,*y*) and location of observation *i*. Given that near points to the reference points have more weight than distant points, this study employed a normal distribution as a kernel function. Additionally, after conducting preliminary tests, as in Ouni and Belloumi [22] and Blazquez and Celis [31], a bandwidth of 1500 m and a 100-m cell size were employed in the KDE analysis. As a result of this analysis, clusters with high concentrations of pedestrian crash injury severities and fatalities were obtained for the studied period.

### 4.2. Partial Proportional Odds Model (PPO)

As aforementioned, the PPO model is used for analyzing different pedestrian crash injury severity levels since this model is an effective alternative model for ordinal response data and has the ability to correct for the violated parallel-lines assumption [25,27,55,58]. The PPO model may be specified in terms of the probability of an injury severity level *j* for a pedestrian crash *i* expressed by Equation (2) [68,69].
(2)$$P\left(Y_{i}>j\right)=\frac{\exp(\beta_{1}X_{1i}+\beta_{2}X_{2i}-\alpha_{j})}{1+\exp(\beta_{1}X_{1i}+\beta_{2}X_{2i}-\alpha_{j})}\;\;j=1,\;2,\ldots ,\;M-1$$
where *Y_i_* is the crash injury severity for crash *i*, ***X*_1*i*_** is a vector with the explanatory variables that comply with the parallel-lines assumption, ***X*_2*i*_** is a vector with the explanatory variables that violate the parallel-lines assumption, ***β*_1_** and ***β*_2_** are vectors of estimated coefficients for ***X*_1*i*_** and ***X*_2*i*_**, respectively, ***α_i_*** is the cut-off threshold of the *j*th cumulative logit, and *M* is the number of injury severity levels (in this study, *M* = 3).

Direct pseudo-elasticities are employed to explain the marginal effect of the explanatory variables on the probability of a pedestrian crash severity level *j* for crash *i*. In this study, the explanatory variables are all binaries, and, thus, the elasticity may not be calculated since it is not differentiable. Instead, pseudo-elasticities are computed as the marginal percentage change of the pedestrian crash injury probability when a binary explanatory variable switches from 0 to 1, or from 1 to 0 using Equation (3) [27,70].
(3)$$E_{X_{jik}}^{P\left(Y_{i}>j\right)}=\frac{P\left(Y_{i}>j\right)\left[Given\;X_{jik}=1\right]-P\left(Y_{i}>j\right)\left[Given\;X_{jik}=0\right]}{P\left(Y_{i}>j\right)\left[Given\;X_{jik}=0\right]}$$
where *P*(*Y_i_* > *j*) is computed with Equation (2) and *X_jik_* is the *k*th binary explanatory variable associated with the injury severity level *j* for pedestrian crash *i*. Note that the direct pseudo-elasticity is computed for each injury severity level *j* of crash *i*. Therefore, average direct pseudo-elasticities are obtained for each injury severity level *j* given known elasticities for all crashes [47]. All statistical analyses were performed with a user-written program in STATA 16 [69].

## 5. Results

### 5.1. KDE Analysis Results

Figure 2 shows that a single critical zone with high-risk of crash fatality/severe injury was identified using the KDE analysis. This zone with 659 pedestrian crashes (15.6% of the whole dataset) that is located in the heart of the city mainly includes the municipality of Santiago and some areas of the neighboring municipalities of Estación Central, Independencia, Las Condes, Providencia, Ñuñoa, and Recoleta. Table 3 shows the distribution of pedestrian crashes across different injury severity levels for each municipality within the identified spatial clusters. The municipality of Santiago presents the highest number of pedestrian crashes among all municipalities with 322 crashes (36.7%) during the studied period, followed by Providencia (19.3%) and Estación Central (11.1%). The highest shares of fatal crashes are observed in Estación Central (24.7%) and Independencia (22.2%), while the highest proportions of serious injuries are concentrated in Ñuñoa (81.1%) and Santiago (75.8%).

The descriptive statistics of the analyzed variables and their pedestrian crash severity outcomes for the identified critical zone are presented in Table 4. In this critical zone, the proportion of serious injuries is higher than for the whole dataset (70.7%), while it is lower for less serious and fatal severities (19.3% and 10%, respectively). The pedestrian crashes in the critical zone are characterized by a higher proportion of male (59.2%) and adult (56.6%) victims than the whole dataset. Regarding the built environment characteristics, a lower proportion of crashes along straight road sections (37.3%) and higher share of crashes at intersections with functioning traffic lights (48.6%) are observed within the critical zone than the whole dataset in Santiago. The distribution of pedestrian crashes among different times of the day, days of the week, and seasons of the year is similar to the statistics for the whole dataset. Most pedestrian crashes in the critical zone are caused by the imprudence of the pedestrian (35.1%), followed by the imprudence of the driver (22.2%) and undetermined causes (15.6%). Regarding the SES, the larger proportions of pedestrian crashes tend to occur in medium and medium-high SES (36.7% and 42%, respectively) within the critical zone, compared to the whole dataset where they are found mostly in medium-low and medium SES areas (33% and 31.7%, respectively). While most pedestrian crashes tend to arise in commercial and residential areas of Santiago, the proportions of these crashes in the critical zone are similar among areas with commerce, residence, and office space. In both the whole dataset and the critical zone, the highest share of pedestrian crashes is apparent in areas with low population exposure (41.9% and 36.5%, respectively). While pedestrian crashes in the city of Santiago tend to occur away from subway stations, approximately 96% of the pedestrian crashes in the critical zone occurred within short and medium distances from subway stations. Finally, most crashes in both the whole dataset and the identified critical zone tend to arise more than 25 m away from traffic lights, and short distances from bus stops and intersections.

### 5.2. Modeling Results

In this study, the explanatory variables were progressively incorporated following an iterative process until the best model was obtained, as the violation of the parallel-lines assumption was tested using Wald chi-square. The following sub-sections present the model results for the pedestrian crashes in the entire city (whole dataset) and within the identified critical zone, and the comparison between the modeling results. Note that we explored the explanatory variables in the model that may be a source of unobserved heterogeneity (i.e., some factors that affect injury severity in pedestrian crashes remain unknown), which may result in bias and inconsistent coefficient estimates. This unobserved heterogeneity is taken into account in random effects models. The relative variability that is explained by the random effects in the model was evaluated with the intra-class correlation coefficient (ICC) for the whole dataset and within the identified critical zone [19,45,59].

#### 5.2.1. Whole Dataset

Table 5 presents the statistically significant parameter estimates for the best model when considering the dataset as a whole. Results show that the ICC values are less than 1% for this model, which suggests that random effects are not significant in the model. The estimates of the PPO model thresholds between the injury severity levels are shown in this table along with those variables that are found to violate the parallel-lines assumption. Since the direction of the effect of the explanatory variables is not always determined by the sign of the coefficients, the average direct pseudo-elasticities for the statistically significant explanatory variables of the PPO model are used to better interpret the results, as shown in Table 6. Note that the summation of the average direct pseudo-elasticities across all severity levels is equal to zero since any increase in the probability of a severity level must be balanced by a decrease in another severity level [27,70,71].

Regarding the demographic characteristics, a significant difference exists between the elderly over 65 years old and children less than 18 years old. The probability of being killed in a pedestrian crash for the elderly increases by 7.13% when contrasted with children less than 18 years old. Compared to female pedestrians, the probabilities of male pedestrians incurring serious injuries or death are increased by 1.93% and 1.73%, respectively, while the probability of less serious injuries decreases by 3.66%. With respect to the relative location of pedestrian crashes, there is only a significant difference between pedestrian crashes at intersections with functioning traffic lights and straight road segments. Pedestrian crashes at intersections with functioning traffic lights could increase the probability of serious injuries and fatalities by 2.05% and 2.16%, respectively, compared to pedestrian crashes that occur along straight road sections.

Model findings reveal that time is significantly associated with injury severities. Pedestrian crashes in the morning, afternoon, and night have comparable increases in less serious injury outcome probabilities with 4.79%, 4.90%, and 4.64%, respectively, while pedestrian crashes during these same time periods seem to be more likely to reduce serious injuries compared to the early morning. The day of the week was found as another significant parameter affecting injury severity of pedestrians. The probability of occurrence of serious and fatal severity increases by 1.35% and 1.42%, respectively, and the probability of less serious injuries decreases by 2.78% during the weekend, compared to a weekday. The model results did not find any significant difference in the probabilities of the three injury levels when discerning between weekdays and holidays, and also among the different seasons of the year.

Among all the significant variables of the model, the contributing factors of pedestrian crashes present the highest increases in the probability of fatality risk. Disobeying traffic signage, driving under the influence of alcohol, pedestrians being under the influence of alcohol, and imprudence of the pedestrian increase the probability of fatal outcome by 17.22%, 12.54%, 10.85%, and 7.78%, respectively, when compared to the imprudence of the driver, while undetermined causes and other causes are linked to decreases in fatality probabilities (−9.55% and −5.88%, respectively). No significant differences were observed between the pedestrian being under the influence of alcohol, speeding, and loss of control of the vehicle, and the base variable for the contributing factor imprudence of the driver.

Regarding the effect of built environment factors, the model results revealed that there is no significant difference between medium-low SES and low SES. However, pedestrian crashes that occur in medium and medium-high SES are associated with decreases in serious injury probabilities (−4.95% and −5.65%, respectively) and comparable decreases in fatality probabilities (−3.72% and −3.59%, respectively). Finally, the probability of occurrence of fatalities increases by 2.17% and 5.48% for medium and long distances from intersections, respectively, when compared to pedestrian crashes that occur near to intersections. Note that the findings did not uncover any significant differences in the injury or fatality probabilities between different distances from bus stops, subway stations, and traffic lights to pedestrian crash locations. In addition, all types of land use and different population exposure levels did not have significantly different effects.

#### 5.2.2. Pedestrian Crashes within the Identified Critical Zone

Results for the analysis of the risk factors that affect crash injury severity in the identified critical zone using the PPO model are shown in Table 7. Note that the best model found has statistically significant parameter estimates only for certain variables related to crash and built environment characteristics, and null associations are observed with the remaining variables. The ICC value is less than 1% for this PPO model, indicating that random effects are not significant. Table 8 presents the average pseudo-elasticities for the three injury severity levels within the high-risk area with pedestrian crash injury severity. When compared to the imprudence of the driver, this table indicates that drunk driving is a relevant contributing factor among the crash characteristics since it is associated with a major increase in the fatality probability (70.75%), followed by signage disobedience (19.52%), and imprudence of the pedestrian (17.88%). The model results also provided evidence of the effect of built characteristics. Pedestrian crashes that occurred at a distance from traffic lights present an increase in the serious injury probability (4.35%) and a non-significant increase in the fatality probability (2.56%), compared to crashes at a short distance from traffic lights. Finally, a decrease is perceived in the fatality probability (−2.39%) for pedestrian crashes that occur within medium distances from intersections, when compared to short distances from intersections.

#### 5.2.3. Comparison of the Modeling Results

When comparing the modeling results for the whole dataset and the identified critical zone, differences and similarities are observed between the explanatory variables in the models. This finding confirms the need to conduct a spatial analysis with the KDE method to determine the variables that may have a significant impact on pedestrian injury severity within high-risk areas.

Higher increases in the fatality probability are perceived within the critical zone for statistically significant contributing factors, suggesting that pedestrians are at a higher risk of being killed in this zone than in other areas of the city due to drunk driving, signage disobedience, and imprudence of the pedestrian. When compared to the imprudence of the driver, speeding is a contributing factor that substantially reduces the probability in serious injury outcomes in the identified critical zone (−74.19%) but has a non-significant effect for the dataset as a whole. For the built environment factors, there is a statistically significant rise in serious injuries for pedestrian crashes at a long distance from traffic lights within the critical zone and a non-significant variation in this probability for the whole dataset. Lastly, intersections are built environment characteristics that became relevant in both model results, but with contrary effects. Pedestrian crashes that occur within the spatial cluster decrease the probability of fatal severity by 2.39% for medium distances from intersections, while this probability increases by 2.17% when considering the whole dataset.

## 6. Discussion

### 6.1. Demographics

The model results for the whole dataset reveal that pedestrians over the age of 65 years old have a higher risk of being involved in a crash than child pedestrians, similar to the recent work conducted in Chile by Bravo Rojas et al. [72]. These results also concur with several studies that have reported the effect of older pedestrians on suffering more severe injuries or death than other age groups [15,16,23,30,32,54]. Elderly individuals have restricted cognitive abilities, less mobility, and lower walking speed than pedestrians from other age groups, which increases the chances of being involved in a crash resulting in some type of injury. Different studies have concluded that most fatal pedestrian victims are predominantly males [34,43,50,54]. Male pedestrians take more risks and are less cautious than female pedestrians, and, thus, they are more likely to be involved in a crash.

### 6.2. Crash Characteristics

With respect to the relative location of crashes, the model results suggest that intersections with functioning traffic lights increase the chances of being killed and seriously injured, which conforms to the large number of pedestrian crashes that occurred during the studied period at signalized intersections in Santiago. In terms of the time of day, the findings concur with the studies by Kim et al. [16] and Chen and Fan [49], indicating that pedestrian crashes in the afternoon tend to decrease the chances of fatal outcomes since there are still some daylight conditions and more traffic congestion with overall low vehicle speeds, whereas pedestrian crashes at night tend to increase fatality probability, as found in various previous studies by Martínez-Ruiz et al. [20], Song et al. [44], and Chen and Fan [49], denoting a dangerous time period for pedestrians due to poor visibility or fatigue. The model results for the whole dataset indicate that pedestrian crashes during the weekend increase the likelihood of serious injuries and fatal crashes, similar to the results of Chen and Fan [49]. The traffic volume is reduced during weekends, and as a result, vehicles travel at higher speeds increasing the odds of fatal pedestrian crashes.

Overall, this study suggests multiple contributing factors for pedestrian safety improvements. Among all significant contributing factors, signage disobedience leads to the highest rise in fatality probability of pedestrian crashes in Santiago. This risky illegal behavior is immersed in the Chilean culture that has been adopted by the population over the years from a young age, perhaps due the lack of enforcement of traffic regulations that creates a general perception that the traffic regulations do not apply to pedestrians [73]. In accordance with Li and Fan [17] and Downey et al. [36], our results reveal that the imprudence of the pedestrian presents important increments of fatal risk. Crossing the road without adequate attention to traffic conditions or standing or walking on the roadway instead of sidewalks are common and recurrent behaviors among Chilean pedestrians [74]. Thus, protective measures should consider reduction of vehicle speed limits, traffic calming, and local area traffic management especially in zones with large flow of pedestrians [75]. Consistent with studies by Kim et al. [16] and Li and Fan [19], our results also reveal that driving under the influence of alcohol presents a significant increase in the probability of being killed in a pedestrian crash when analyzing such crashes using the whole dataset. Additionally, studies have found that pedestrians being under the influence of alcohol is a crucial concern [26,33,34]. The work by Sun et al. [26] found that pedestrians with alcohol involvement are 180% more likely to be killed or severely injured. In this study, crashes that involve pedestrians under the influence of the alcohol represent only 3% of all pedestrian crashes, however there is a significant increase in the probability of crash fatal severity for this contributing cause when focusing on the whole dataset. This result is particularly interesting since previous studies have identified that legislation enacted in Chile during the analyzed period for increasing penalties for drivers under the influence of alcohol and decreasing the maximum blood alcohol limit from 0.05 to 0.03 g/dL was associated with considerable reduction in alcohol-related traffic fatalities and injuries [76,77]. However, these studies only focus on overall traffic crashes rather than on pedestrian crashes, which suggests that these laws may have been particularly relevant in targeting a specific group of drivers rather than protecting pedestrians.

### 6.3. Built Environment Characteristics

The city of Santiago has undergone major investments in road infrastructure during the last decades including improvements in traffic signage, roundabouts, pedestrian crossings, and exclusive corridors for public transportation [10]. Apparently, these improvements offer safer mobility and visibility of vulnerable road users, such as pedestrians, in more advantaged neighborhoods than in deprived neighborhoods [41]. These results assist road safety professionals in implementing adequate safety measures and improving pedestrian infrastructure in low SES zones of the city to reduce pedestrian crashes.

Distances between pedestrian crash locations and intersections greater than 10 m tend to increase the odds of fatality outcomes, meaning that risk propensity for fatal injuries is increased away from intersections, coinciding with the works by Mukherjee and Mitra [5] and Pour et al. [32]. Pedestrians may start jaywalking as they get closer to intersections, and, thus, the introduction of pedestrian crossings at mid-blocks may help improve pedestrian safety.

### 6.4. Model Comparison

The KDE analysis results indicate that a single critical zone with high concentrations of pedestrian crashes was identified comprising 7 out of 34 municipalities of the city. This critical zone is situated the heart of the city, particularly in the municipality of Santiago, in which the abundance of economic, cultural, and political activities attracts a daily floating population of one third of the total population of Santiago [78]. Consequently, pedestrians are more prone to be exposed to injury severity crashes in this municipality. In addition, a high density of pedestrian crash fatalities is observed in the municipalities of Estación Central and Providencia, which have experienced a rapid redevelopment of high-rise properties and an attraction of a large number of pedestrians to various shopping malls and other commercial spaces.

The model results show that common risk factors are statistically significant in pedestrian crash models for both the whole crash dataset in the city and within the identified critical zone. The contributing factors related to the imprudence of the pedestrian, driving under the influence of alcohol, and signage disobedience present considerably higher probabilities of pedestrian crash fatalities in the critical zone than in the entire city. The high concentration of alcohol outlets in the critical area may contribute to the pedestrian crash occurrence while the driver is under the influence of alcohol [14], and the high pedestrian volume in this critical area may produce more jaywalking, careless actions of pedestrians, and violation of traffic signage than in the rest of the city.

When focusing on the whole dataset, the probability of fatal pedestrian crashes increases for medium and long distances from intersections whereas there is a reduction in the probability of suffering severe injuries for medium distances from intersections in critical zones with high density of crash injury severity clusters. These contrasting results between the two models should be further investigated since the crash database has no specific information on the presence/absence of traffic lights, pedestrian crossings, or other signage at these intersections.

## 7. Conclusions

This research aimed to investigate the risk factors that significantly impact pedestrian crash injury severity by performing spatial and statistical analyses. Based on pedestrian crash data collected between 2012 and 2016 in Santiago, Chile, a KDE analysis was first performed to identify spatial clusters with high density of pedestrian crashes, and, subsequently, PPO models were built to understand different risk factors related to the pedestrian, crash, and built environment characteristics that significantly affect injury severity levels of pedestrian crashes within the identified spatial cluster and also for the whole city. Results for the whole dataset confirm previous identified risk factors that play relevant roles in increasing the probability of serious injury and fatality outcomes in pedestrian crashes. For example, the following significantly increase the odds of resulting in a fatal crash: being male and older, being the morning, night, or a weekend, being at a distance from intersections with functioning traffic lights, the presence of alcohol in the driver or the pedestrian, imprudence of the pedestrian, and signage disobedience. Safe mobility for the elderly should be ensured by providing traffic regulations related to pedestrians’ right-of-way and crossing times [41]. Additionally, education programs targeted particularly at men are necessary to help their awareness on safe road crossing practices. With respect to crashes during the night, pedestrians need crossing signs equipped with flashing lights to alert drivers of the presence of pedestrians.

Common risk factors for the whole dataset and the identified critical area were identified to increase the probability of pedestrian crash injury severity. However, significantly higher increases are perceived in the probability of fatality risk for certain contributing factors in pedestrian crashes that occur within the critical zone. This result proves the need to perform a spatial clustering analysis with the KDE technique to identify those risk factors that require effective pedestrian safety improvements and interventions to increase pedestrian safety in high-risk zones.

In light of our results, we provide the following set of countermeasures as practical recommendations that may be introduced in Santiago for improving pedestrian safety: improving traffic regulations to secure right-of-way of pedestrians; introducing devices that favor the significant increasing of crossing times; crossing signs equipped with flashing lights at night to alert drivers to the presence of pedestrians; implementing speed cameras in critical zones; and designing and implementing road safety education programs to teach both new drivers and pedestrians to respect traffic regulations. In this study, the low ICC values obtained for each model reveal the insignificance of the random effects in the models. Therefore, the studied variables were not found to have random effects on the probability of pedestrian crash injury severities. However, future research should compare our results with other approaches such as the mixed effect PPO model to better explore the unobserved heterogeneity in the crash data. Further research should also consider other risk factors such as weather, road characteristics, and vehicle conditions that may have significant impacts on pedestrian crash injury severity, similar to the works by Munira et al. [21], Pour-Rouholamin and Zhou [24], and Park and Bae [79]. The model may improve the explanation of the data if other additional explanatory variables or proxy variables are available. As aforementioned, this study has the limitation of using surrogate data for population exposure since actual pedestrian counts are not available. In future research, data collection methods should be studied to obtain pedestrian counts throughout the city to be employed as a risk factor in pedestrian crash injury severity modeling. The analysis of pedestrian crashes using actual pedestrian volume counts would provide more accurate modeling results.

This study has the limitation of using pedestrian crash data between 2012 and 2016 due to main three reasons. First, recent crash data were not included in the study since we considered that the social riots that took place in Chile in 2019 and the pandemic that started in March 2020 may have influenced our results, given this atypical situation. Second, the pedestrian crash data were available for this period when this study was started. Third, Iglesias et al. [10] indicated that during the 2010–2016 period a total of USD 3.8 million was invested in transportation infrastructure projects in Santiago. In addition, Martínez et al. [41] studied the improvements in Santiago that offer safer mobility and visibility of pedestrians over a 12-year period (2002–2013). Thus, this study analyzes a five-year period of pedestrian crashes when these improvements were being carried out. We assume that such crashes reflect the situation with safety improvements. Future research will also determine whether certain risk factors that affect pedestrian crashes persist through time by analyzing the models per year, and, thus, adequate countermeasures may be implemented to directly target those factors.

## Figures and Tables

**Figure 1 ijerph-19-11126-f001:**
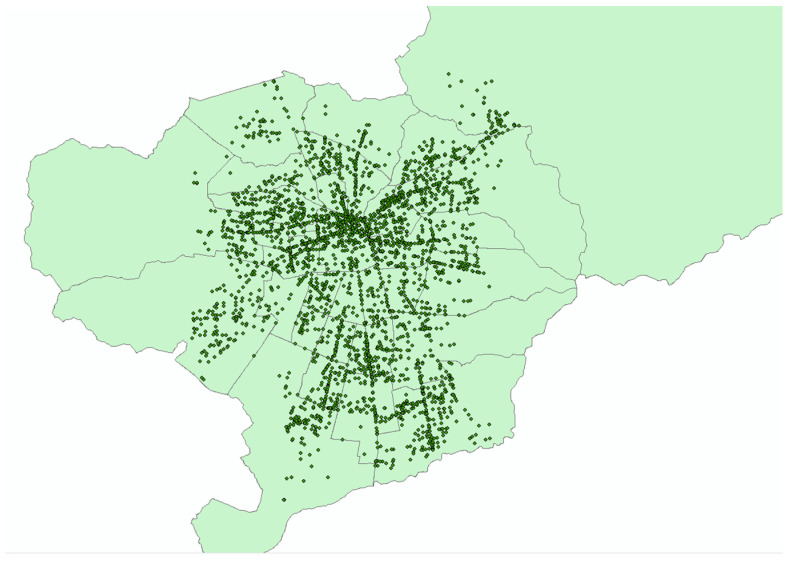
Pedestrian crashes with injury severity in Santiago during the 2012–2016 period, Source: Prepared by the authors.

**Figure 2 ijerph-19-11126-f002:**
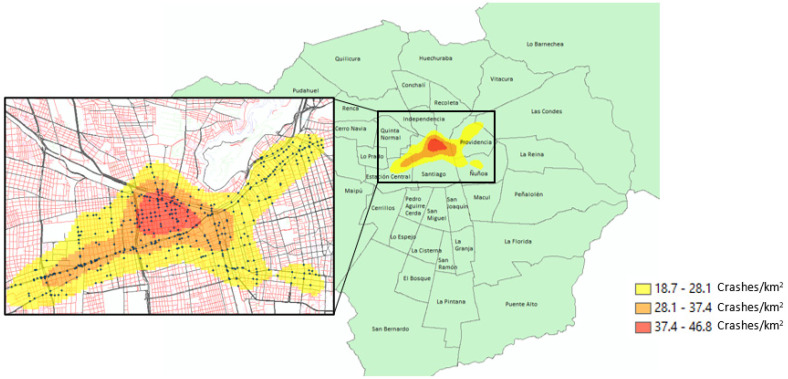
Critical zone with high density of pedestrian crashes during the 2012–2016 period.

**Table 1 ijerph-19-11126-t001:** Description of pedestrian crash studies using KDE and statistical methods.

Authors	Year	Study Description
Xie et al. [65]	2017	The authors computed pedestrian crash costs weighted by injury severity using KDE, and employed a tobit model to relate the contributing factors to the crash costs.
Chimba et al. [62]	2018	The authors identified high concentrations of pedestrian crashes in Tennessee using KDE and applied a negative binomial to test the statistical significance of explanatory variables related to sociodemographic characteristics.
Ouni & Belloumi [22]	2018	The authors implemented an integrated two-step approach by first identifying spatial clusters of vulnerable road user (e.g., pedestrians) crashes, and second, assessing the influence of personal and environmental factors on injury severity in Tunisia using multinomial logit models.
Yao et al. [63]	2018	KDE was applied to estimate pedestrian crash density in Shanghai, China, and then the random forest method was employed for modeling pedestrian crashes.
Hu et al. [15]	2020	KDE was utilized to identify clusters of pedestrian crashes and other variables, and subsequently explored the relationship and interaction between building environment characteristics and pedestrian injury risk using binary logistic regression and tree-based models.
Bajada & Attard [43]	2021	A statistical analysis was performed using multivariate techniques to investigate the association between variables related to crash characteristics and pedestrian fatalities and injuries in Malta, and, subsequently, KDE was used to identify high-risk areas with increased likelihood for pedestrian injury crashes.
Chen et al. [64]	2022	The authors introduced the geographically and temporally weighted ordered logistic regression model integrated with KDE to model pedestrian crash severity in rural highways of the Anhui Province, China.

**Table 2 ijerph-19-11126-t002:** Descriptive statistics of variables and their pedestrian crash severity outcomes for the whole dataset.

Variable		Frequency		Injury Severity (%)		
				Less Serious	Serious	Fatal
Pedestrian crashes		4216	(100.0%)	22.2	64.8	13.0
*Demographics*						
Age ^a^	Child (<18 years old)	666	(15.8%)	24.3	66.4	9.3
	Young adult (18–24 years old)	412	(9.8%)	28.4	62.4	9.2
	Adult (25–65 years old)	2081	(49.4%)	22.0	65.4	12.6
	Elderly (>65 years old)	1057	(25.1%)	18.8	63.7	17.5
Gender ^a^	Female	1812	(43.0%)	24.1	66.1	9.8
	Male	2404	(57.0%)	20.8	63.9	15.3
*Crash characteristics*						
Relative location ^a^	Straight road section	1775	(42.1%)	22.7	62.8	14.5
	Curved road section	10	(0.2%)	20.0	70.0	10.0
	Intersection without signage	142	(3.4%)	17.6	70.4	12.0
	Intersection with functioning traffic lights	1420	(33.7%)	18.9	67.2	13.9
	Intersection with yield sign	220	(5.2%)	24.5	68.6	6.8
	Intersection with stop sign	184	(4.4%)	31.0	60.3	8.7
	Sidewalk or shoulder	50	(1.2%)	30.0	56.0	14.0
	Disabled access	30	(0.7%)	10.0	73.3	16.7
	Other	385	(9.1%)	28.1	63.9	8.1
Time ^a^	Early morning	702	(16.7%)	19.7	68.5	11.8
	Morning	998	(23.7%)	22.8	64.1	13.0
	Afternoon	1101	(26.1%)	22.9	67.7	9.4
	Night	1415	(33.6%)	22.5	61.3	16.3
Day ^a^	Weekday	3054	(72.4%)	22.7	65.1	12.2
	Weekend	1054	(25.0%)	20.6	63.8	15.7
	Holiday	108	(2.6%)	25.0	67.6	7.4
Season ^a^	Fall	1115	(26.4%)	22.5	65.7	11.8
	Winter	1190	(28.2%)	22.7	63.7	13.6
	Spring	1050	(24.9%)	21.4	66.0	12.6
	Summer	861	(20.4%)	22.1	63.9	14.1
Contributing cause ^a^	Imprudence of driver	1280	(30.4%)	26.4	65.1	8.5
	Imprudence of pedestrian	1436	(34.1%)	17.3	62.6	20.1
	Driving under the influence of alcohol	74	(1.8%)	35.1	40.5	24.3
	Pedestrian under the influence of alcohol	125	(3.0%)	18.4	58.4	23.2
	Speeding	28	(0.7%)	28.6	50.0	21.4
	Loss of control of vehicle	60	(1.4%)	16.7	71.7	11.7
	Signage disobedience	249	(5.9%)	21.7	51.4	26.9
	Undetermined causes	527	(12.5%)	23.7	72.8	3.5
	Other causes	437	(10.4%)	24.1	74.6	1.3
*Built environment characteristics*						
Socioeconomic status ^b^	Low	150	(3.6%)	16.0	64.0	20.0
	Medium-low	1391	(33.0%)	21.5	63.6	14.9
	Medium	1337	(31.7%)	23.8	62.5	13.7
	Medium-high	741	(17.6%)	22.1	67.2	10.7
	High	597	(14.2%)	21.9	70.0	8.0
Land use ^b^	Commercial (m^2^)	3451	(33.9%)	21.6	66.0	12.4
	Industrial (m^2^)	1611	(15.8%)	22.1	65.1	12.8
	Office space (m^2^)	1766	(17.4%)	21.8	66.0	12.1
	Residential (m^2^)	3339	(32.8%)	20.8	68.4	10.8
Population exposure ^b^	Low (0–500 inhabitants)	1539	(36.5%)	22.2	62.6	15.3
	Medium (500–1000 inhabitants)	1423	(33.8%)	22.6	65.8	11.6
	High (≥1000 inhabitants)	1254	(29.7%)	21.8	66.5	11.7
Bus stops ^c^	Short distance (0–100 m)	3493	(82.9%)	21.9	64.8	13.3
	Medium distance (100–250 m)	634	(15.0%)	22.7	66.1	11.2
	Long distance (≥250 m)	89	(2.1%)	29.2	57.3	13.5
Subway stations ^c^	Short distance (0–250 m)	713	(16.9%)	21.5	64.2	14.3
	Medium distance (250–1000 m)	1161	(27.5%)	20.3	67.7	12.0
	Long distance (≥1000 m)	2342	(55.6%)	23.4	63.6	13.1
Traffic lights ^d^	Short distance (0–10 m)	812	(19.3%)	20.7	68.1	11.2
	Medium distance (10–25 m)	651	(15.4%)	21.8	65.6	12.6
	Long distance (≥25 m)	2753	(65.3%)	22.7	63.7	13.6
Intersections ^d^	Short distance (0–10 m)	2158	(51.2%)	23.1	66.1	10.8
	Medium distance (10–25 m)	1256	(29.8%)	22.8	63.9	13.3
	Long distance (≥25 m)	802	(19.0%)	19.0	62.8	18.2

Data source: ^a^ Chilean National Road Safety Commission (CONASET), https://mapas-conaset.opendata.arcgis.com/ (accessed on 17 April 2018); ^b^ National Statistics Institute (INE), https://www.censo2017.cl/ (accessed on 8 June 2019); ^c^ Spatial Data Infrastructure-City Observatory (IDE-OC), https://ideocuc-ocuc.hub.arcgis.com/datasets/ (accessed on 10 June 2019); ^d^ Center for Sustainable Urban Development (CEDEUS), http://datos.cedeus.cl/layers/ (accessed on 10 June 2019).

**Table 3 ijerph-19-11126-t003:** Pedestrian crashes and injury severity per municipality within the identified critical zone.

Municipality	Frequency		Injury Severity (%)		
			Less Serious	Serious	Fatal
Estación Central	97	(11.1%)	22.7	52.6	24.7
Independencia	18	(2.1%)	0.0	77.8	22.2
Las Condes	5	(0.6%)	60.0	40.0	0.0
Providencia	169	(19.3%)	26.7	68.0	5.3
Ñuñoa	53	(6.0%)	15.1	81.1	3.8
Recoleta	18	(2.1%)	16.7	66.6	16.7
Santiago	322	(36.7%)	16.1	75.8	8.1

**Table 4 ijerph-19-11126-t004:** Descriptive statistics of variables and their pedestrian crash severity outcomes for the identified critical zone.

Variable		Frequency				Injury Severity (%)		
		Whole Dataset		Critical Zone		Less Serious	Serious	Fatal
Pedestrian crashes		4216	(100.0%)	659	(100%)	19.3	70.7	10.0
*Demographics*								
Age	Child (<18 years old)	666	(15.8%)	44	(6.7%)	13.6	79.5	6.8
	Young adult (18–24 years old)	412	(9.8%)	74	(11.2%)	24.3	67.6	8.1
	Adult (25–65 years old)	2081	(49.4%)	373	(56.6%)	19.8	72.1	8.0
	Elderly (>65 years old)	1057	(25.1%)	168	(25.5%)	17.3	66.7	16.1
Gender	Female	1812	(43.0%)	269	(40.8%)	23.1	71.0	5.9
	Male	2404	(57.0%)	390	(59.2%)	16.7	70.5	12.8
*Crash characteristics*								
Relative location	Straight road section	1775	(42.1%)	246	(37.3%)	19.9	69.1	11.0
	Curved road section	10	(0.2%)	2	(0.3%)	50.	50.0	0.0
	Intersection without signage	142	(3.4%)	7	(1.1%)	14.3	85.7	0.0
	Intersection with functioning traffic lights	1420	(33.7%)	320	(48.6%)	17.2	72.5	10.3
	Intersection with yield sign	220	(5.2%)	25	(3.8%)	28.0	72.0	0.0
	Intersection with stop sign	184	(4.4%)	9	(1.4%)	33.3	55.6	11.1
	Sidewalk or shoulder	50	(1.2%)	15	(2.3%)	20.0	73.3	6.7
	Disabled access	30	(0.7%)	6	(0.9%)	0.0	83.3	16.7
	Other	385	(9.1%)	29	(4.4%)	27.6	62.1	10.3
Time	Early morning	702	(16.7%)	102	(15.5%)	17.6	73.5	8.8
	Morning	998	(23.7%)	169	(25.6%)	22.5	67.5	10.1
	Afternoon	1101	(26.1%)	151	(22.9%)	19.2	74.2	6.6
	Night	1415	(33.6%)	237	(36.0%)	17.7	69.6	12.7
Day	Weekday	3054	(72.4%)	492	(74.7%)	18.9	72.2	8.9
	Weekend	1054	(25.0%)	155	(23.5%)	21.3	65.2	13.5
	Holiday	108	(2.6%)	12	(1.8%)	8.3	83.3	8.3
Season	Fall	1115	(26.4%)	178	(27.0%)	21.9	67.4	10.7
	Winter	1190	(28.2%)	191	(29.0%)	16.8	72.3	11.0
	Spring	1050	(24.9%)	151	(22.9%)	16.6	73.5	9.9
	Summer	861	(20.4%)	139	(21.1%)	22.3	69.8	7.9
Contributing factor	Imprudence of driver	1280	(30.4%)	146	(22.2%)	25.4	71.2	3.4
	Imprudence of pedestrian	1436	(34.1%)	231	(35.1%)	13.0	67.5	19.5
	Driving under the influence of alcohol	74	(1.8%)	5	(0.8%)	0.0	40.0	60.0
	Pedestrian under the influence of alcohol	125	(3.0%)	18	(2.7%)	11.1	77.8	11.1
	Speeding	28	(0.7%)	4	(0.6%)	50.0	25.0	25.0
	Loss of control of vehicle	60	(1.4%)	8	(1.2%)	25.0	75.0	0.0
	Signage disobedience	249	(5.9%)	60	(9.1%)	23.3	63.4	13.3
	Undetermined causes	527	(12.5%)	103	(15.6%)	22.3	77.7	0.0
	Other causes	437	(10.4%)	84	(12.7%)	20.2	77.4	2.4
*Built environment characteristics*								
Socioeconomic status	Low	150	(3.6%)	7	(1.1%)	14.3	71.4	14.2
	Medium-low	1391	(33.0%)	79	(12.0%)	17.7	63.3	19.0
	Medium	1337	(31.7%)	242	(36.7%)	17.4	69.4	13.2
	Medium-high	741	(17.6%)	277	(42.0%)	20.9	72.6	6.5
	High	597	(14.2%)	54	(8.2%)	22.2	77.8	0.0
Land use	Commercial (m^2^)	3451	(33.9%)	632	(28.8%)	19.8	70.6	9.6
	Industrial (m^2^)	1611	(15.8%)	367	(16.7%)	21.2	67.6	11.2
	Office space (m^2^)	1766	(17.4%)	587	(27.7%)	19.1	71.5	9.4
	Residential (m^2^)	3339	(32.8%)	607	(26.8%)	19.6	70.8	9.6
Population exposure	Low (0–500 inhabitants)	1539	(36.5%)	276	(41.9%)	18.1	68.1	13.8
	Medium (500–1000 inhabitants)	1423	(33.8%)	163	(24.7%)	22.1	70.6	7.4
	High (≥1000 inhabitants)	1254	(29.7%)	220	(33.4%)	18.6	74.1	7.3
Bus stops	Short distance (0–100 m)	3493	(82.9%)	572	(86.8%))	18.5	71.2	10.3
	Medium distance (100–250 m)	634	(15.0%)	87	(13.2%)	24.1	67.8	8.1
	Long distance (≥250 m)	89	(2.1%)	0	(0.0%)	0.0	0.0	0.0
Subway stations	Short distance (0–250 m)	713	(16.9%)	303	(46.0%)	19.2	67.3	13.5
	Medium distance (250–1000 m)	1161	(27.5%)	329	(49.9%)	20.1	72.3	7.6
	Long distance (≥1000 m)	2342	(55.6%)	27	(4.1%)	11.1	88.9	0.0
Traffic lights	Short distance (0–10 m)	812	(19.3%)	256	(38.8%)	20.7	69.9	9.4
	Medium distance (10–25 m)	651	(15.4%)	121	(18.4%)	22.3	66.1	11.6
	Long distance (≥25 m)	2753	(65.3%)	282	(42.8%)	16.7	73.4	9.9
Intersections	Short distance (0–10 m)	2158	(51.2%)	381	(57.8%)	17.3	74.0	8.7
	Medium distance (10–25 m)	1256	(29.8%)	166	(25.2%)	24.1	63.2	12.7
	Long distance (≥25 m)	802	(19.0%)	112	(17.0%)	18.8	70.5	10.7

**Table 5 ijerph-19-11126-t005:** Results of the PPO model using the whole dataset.

Variable		Threshold 1: Less Serious vs. Serious, Fatal		Threshold 2: Less Serious, Serious vs. Fatal	
		Coefficients	Standard Error	Coefficients	Standard Error
*Demographics*					
Age	base: Child (<18 years old)				
	Elderly (>65 years old) ^a^	0.375 ***	0.117	0.748 ***	0.129
Gender	base: Female				
	Male	0.216 **	0.068	0.216 **	0.068
*Crash characteristics*					
Relative location	base: Straight road section				
	Intersection with functioning traffic lights	0.257 **	0.088	0.257 **	0.088
Time	base: Early morning				
	Morning ^a^	−0.273 *	0.118	0.099	0.141
	Afternoon	−0.281 **	0.106	−0.281 **	0.106
	Night ^a^	−0.269 *	0.112	0.199	0.128
Day	base: Weekday				
	Weekend	0.169 *	0.077	0.169 *	0.077
Contributing factor	base: Imprudence of driver				
	Imprudence of pedestrian ^a^	0.474 ***	0.099	0.847 ***	0.124
	Driving under the influence of alcohol ^a^	−0.486	0.260	1.035 ***	0.299
	Pedestrian under the influence of alcohol ^a^	0.319	0.246	0.936 ***	0.244
	Speeding ^a^	−0.061	0.429	1.115 *	0.480
	Signage disobedience ^a^	0.189	0.171	1.331 ***	0.180
	Undetermined causes ^a^	0.086	0.122	−2.020 ***	0.395
	Other causes ^a^	0.111	0.133	−0.983 ***	0.276
*Built environment characteristics*					
Socioeconomic status	base: Low				
	Medium	−0.491 **	0.184	−0.491 **	0.184
	Medium-high	−0.503 **	0.195	−0.503 **	0.195
	High ^a^	−0.274	0.207	−0.777 ***	0.239
Intersections	base: Short distance (0–10 m)				
	Medium distance (10–25 m) ^a^	0.002	0.091	0.255 *	0.116
	Long distance (≥25 m) ^a^	0.224 *	0.112	0.580 ***	0.127
*Cut-off points*		1.219 ***	0.270	−2.714 ***	0.288
*Number of observations*		4216			
*Log likelihood*		−3463.87			
*Pseudo R^2^*		0.1665			

^a^ Parallel-lines assumption is violated; * *p* < 0.05, ** *p* < 0.01, *** *p* < 0.001.

**Table 6 ijerph-19-11126-t006:** Average pseudo-elasticities for the PPO model using the whole dataset.

Variable		Average Pseudo-Elasticity (%)		
		Less Serious	Serious	Fatal
*Demographics*				
Age	base: Child (<18 years old)			
	Elderly (>65 years old)	−5.98 ***	−1.15	7.13 ***
Gender	base: Female			
	Male	−3.66 **	1.93 **	1.73 **
*Crash characteristics*				
Relative location	base: Straight road section			
	Intersection with functioning traffic lights	−4.21 **	2.05 **	2.16 **
Time	base: Early morning			
	Morning	4.79 *	−5.61 **	0.83
	Afternoon	4.90 **	−2.74 *	−2.16 **
	Night	4.64 *	−6.30 ***	1.66
Day	base: Weekday			
	Weekend	−2.78 *	1.35 *	1.42 *
Contributing factor	base: Imprudence of driver			
	Imprudence of pedestrian	−7.63 ***	−0.15	7.78 ***
	Driving under the influence of alcohol	9.25	−21.79 ***	12.54 *
	Pedestrian under the influence of alcohol	−4.91	−5.94	10.85 **
	Signage disobedience	−3.03	−14.19 ***	17.22 ***
	Undetermined causes	−1.42	10.97 ***	−9.55 ***
	Other causes	−1.82	7.71 ***	−5.88 ***
*Built environment characteristics*				
Socioeconomic status	base: Low			
	Medium	8.67 *	−4.95 *	−3.72 **
	Medium-high	9.24 *	−5.65 *	−3.59 **
	High	4.86	0.21	−5.07 ***
Intersections	base: Short distance (0–10 m)			
	Medium distance (10–25 m)	−0.04	−2.13	2.17 **
	Long distance (≥25 m)	−3.62 *	−1.87	5.48 ***

* *p* < 0.05, ** *p* < 0.01, *** *p* < 0.001.

**Table 7 ijerph-19-11126-t007:** Results of the PPO model using the pedestrian crashes in the identified critical zone.

Variable		Threshold 1: Less Serious vs. Serious, Fatal		Threshold 2: Less Serious, Serious vs. Fatal	
		Coefficients	Standard Error	Coefficients	Standard Error
*Crash characteristics*					
Contributing factor	base: Imprudence of driver				
	Imprudence of pedestrian ^a^	0.685 *	0.296	2.359 ***	0.436
	Driving under the influence of alcohol	4.077 ***	1.081	4.077 ***	1.081
	Speeding ^a^	−1.153	1.076	3.205 *	1.261
	Signage disobedience ^a^	−0.011	0.395	1.923 ***	0.566
	Other causes ^a^	0.298	0.337	−0.543 *	0.261
*Built environment characteristics*					
Traffic lights	base: Short distance (0–10 m)				
	Long distance (≥25 m)	0.500 *	0.249	0.500 *	0.249
Intersections	base: Short distance (0–10 m)				
	Medium distance (10–25 m)	−0.543 *	0.261	−0.543 *	0.261
*Cut-off points*		1.199 *	1.068	−4.469 ***	1.128
*Number of observations*		659			
*Log likelihood*		−462.40			
*Pseudo R* ^2^		0.1150			

^a^ Parallel-lines assumption is violated; * *p* < 0.05, *** *p* < 0.001.

**Table 8 ijerph-19-11126-t008:** Average pseudo-elasticities for the PPO model using pedestrian crashes in the identified critical zone.

Variable		Average Pseudo-Elasticity (%)		
		Less Serious	Serious	Fatal
*Crash characteristics*				
Contributing factor	base: Imprudence of driver			
	Imprudence of pedestrian	−9.07 *	−8.81 *	17.88 ***
	Driving under the influence of alcohol	−17.12	−53.63 **	70.75 ***
	Speeding	22.32	−74.19 **	51.87
	Signage disobedience	0.15	−19.68 *	19.52 *
	Other causes	−3.91	2.29	1.62
*Built environment characteristics*				
Traffic lights	base: Short distance (0–10 m)			
	Long distance (≥25 m)	−6.91 *	4.35 *	2.56
Intersections	base: Short distance (0–10 m)			
	Medium distance (10–25 m)	8.37	−5.99	−2.39 *

* *p* < 0.05, ** *p* < 0.01, *** *p* < 0.001.

## Data Availability

The data employed in this study were downloaded from CONASET’s website (https://mapas-conaset.opendata.arcgis.com/ (accessed on 17 April 2018)).

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
