# Peer review of "Investigating the Risk Factors Associated with Injury Severity in Pedestrian Crashes in Santiago, Chile"

_ijerph, 2022, doi:10.3390/ijerph191711126_

Round 1
Reviewer 1 Report
Recommendation
Minor Revision
General comments
The authors deal with the topic of injury risk of pedestrians in Santiago, Chile. They use large dataset of pedestrian crashes during 2012-2016. With the help of statistical tools (KDE, PPO) the authors identify critical zone within Santiago and they also identify risk factors that affect pedestrian crash injury severity.
The study provided by authors has the potential, in my opinion, to have positive impact on the pedestrian safety in Santiago, Chile (if the results are taken into account by stakeholders). Therefore, I am convinced of the social profitability of present study.
I recommend the manuscript for publication after minor revision is done. Please see specific comments.
Specific comments:
Introduction
Line 39: …reported worldwide in 2019 for 22% of all traffic deaths…. This statement is provided with a reference [4], which is in fact 2018 WHO report claiming 26% of all traffic deaths. Please correct either Your statement in manuscript or replace the reference.
Data
Line 153-154: …reported 4,874 pedestrian crashes…of which 4,216 (86.5%)… Please provide reference for these numbers.
Line 157: The Chilean National Road Safety Commision (CONASET) provided the pedestrian crash data…Could You be more specific about the source of the data? Is it a report? If so, please provide reference. Is it a database? If so, please provide link. Details are needed for Your data to be verifiable.
Line 172: Figure 1…The same comment as above. The reference is missing.
Line 202: Table 1. Again the same comment. You claim at lines 204-206 the source as CONASET, INE, IDE-OC, CEDEUS. Could You be more specific? The reader shouldn’t be forced to “google” it but to be provided with direct link or reference.
Results
Line 269: Fig 2. What do particular colors mean (red/orange/yellow)? Please explain.
Conclusion
One of the limitation of the study, in my opinion, is the dataset from the period 2012-2016. Till now, these are 6 to 10 years old data. You claim in the discussion (line 466): The city of Santiago has undergone major investments in road infrastructure during the last decades that include improvements in traffic signage, roundabouts, pedestrian crossings, and exclusive corridors for public transportation [10]. Apparently, these improvements offer safer mobility and visibility of vulnerable road users, such as pedestrians, generating a lower risk of fatal and severe injuries in more advantaged neighborhoods than in deprived neighborhoods [41].
Therefore, my question is, do the data from 2012-2016 still reflect current situation? Please address that in the Conclusion.
Reviewer 2 Report
The authors did a good job on examining the risk factors associated with injury severity in pedestrian crashes in Santiago, Chile. Overall, the paper is well written, understandable, and flows well. The authors clearly explained the background, methodology, and contribution of the study. The findings would be beneficial for practitioners and policymakers in improving pedestrian safety. However, specific comments/concerns to improve the manuscript are offered below:
1. Too long Introduction! Please review your introduction. Perhaps it could be shortened and simplified by removing your review of existing research and methods used, and placing it under a separate heading and thus enabling you to add to it if necessary. Moreover, the information in the literature review section can be summarized in a table.
2. One major concern remains in this paper - it has to do with capturing heterogeneity through applying random parameters. We know that crash is a very complex process in which it is very difficult to capture all the contributory factors during the modeling process. This leads to a very important phenomenon called unobserved heterogeneity. With pedestrian-related crash data, in particular, I would imagine, the sources of heterogeneity will be substantial. Neglecting such effects may result in biases in parameter estimates. Therefore, the authors are suggested to investigate the presence of unobserved heterogeneity in the models.
3. Why did the authors select the specific span of crash data (i.e., 2012-2016)? Why not the most recent ones?
4. Due to the potential temporal instability in crash data, it would be suggested to not use data with a large span. Please explain why you used 5-year period data in your study.
5. Please explain in the manuscript what the authors meant by less serious and serious injury based on the KABCO scale.
6. How did the authors define the ranges used for the predictor variables? For example, short distance (0-10m), medium distance (10-25m), and long-distance (>25 m). More clarifications are needed based on the previous literature or engineering judgement.
7. The Conclusion section is too brief and does not present the practical information of the results that came from the study. I suggest the authors can add an independent section/paragraph discussing safety implications to provide the further application with practical recommendations related to the model estimations.
Reviewer 3 Report
This paper focuses on the analysis of risk factors associated with pedestrians in the city of Santiago, Chile. The work offers a spatial and statistical analysis.
I have some comments for the authors to address as follows.
Acronyms/Abbreviations should be defined the first time they appear in the text (such as OECD on line 45);
On line 191 you separate the statistics related to the driver and pedestrian behavior (influence of alcohol), respectively. When you write “while disobeying traffic signals has the highest share of fatal injury with 26.9%”; is the signage disobedience related to drivers or pedestrians? Furthermore, it might be interesting to separate the percentage related to the imprudence of the driver and the pedestrian (64.5%) as your work is focused on pedestrians.
In Table 1 - crash characteristics section there is a percentage of crashes (1.2%) related to the variable “sidewalk or shoulder”; how did you classify these crashes? Because, maybe, crashes on sidewalks do not imply a conflict between vehicles and pedestrians as sidewalks are the pedestrians’ path; conversely, crashes on the shoulder can involve a conflict with vehicles as this roadway element is closely related to vehicles' movement.
Round 2
Reviewer 2 Report
The authors carefully addressed all my comments and I would be happy to recommend the revised manuscript for publication.